# Coupling between oxygen redox and cation migration explains unusual electrochemistry in lithium-rich layered oxides

William E. Gent [1,2], Kipil Lim [3,4], Yufeng Liang [5], Qinghao Li [2,6], Taylor Barnes [5], Sung-Jin Ahn [7], Kevin H. Stone [4], Mitchell McIntire [8], Jihyun Hong [3,4], Jay Hyok Song [9], Yiyang Li [3], Apurva Mehta [4], Stefano Ermon [8], Tolek Tyliszczak [2], David Kilcoyne [2], David Vine [2], Jin-Hwan Park [7], Seok-Kwang Doo [7], Michael F. Toney [4,10], Wanli Yang [2], David Prendergast [5] & William C. Chueh [3,10]

Lithium-rich layered transition metal oxide positive electrodes offer access to anion redox at high potentials, thereby promising high energy densities for lithium-ion batteries. However, anion redox is also associated with several unfavorable electrochemical properties, such as open-circuit voltage hysteresis. Here we reveal that in $Li_{1.17-x}Ni_{0.21}Co_{0.08}Mn_{0.54}O_2$, these properties arise from a strong coupling between anion redox and cation migration. We combine various X-ray spectroscopic, microscopic, and structural probes to show that partially reversible transition metal migration decreases the potential of the bulk oxygen redox couple by > 1 V, leading to a reordering in the anionic and cationic redox potentials during cycling. First principles calculations show that this is due to the drastic change in the local oxygen coordination environments associated with the transition metal migration. We propose that this mechanism is involved in stabilizing the oxygen redox couple, which we observe spectroscopically to persist for 500 charge/discharge cycles.

[1] Department of Chemistry, Stanford University, 333 Campus Drive, Stanford, CA 94305, USA. [2] The Advanced Light Source, Lawrence Berkeley National Laboratory, 1 Cyclotron Road, Berkeley, CA 94720, USA. [3] Department of Materials Science and Engineering, Stanford University, 496 Lomita Mall, Stanford, CA 94305, USA. [4] Stanford Synchrotron Radiation Lightsource, SLAC National Accelerator Laboratory, 2575 Sand Hill Road, Menlo Park, CA 94025, USA. [5] The Molecular Foundry, Lawrence Berkeley National Laboratory, 1 Cyclotron Road, Berkeley, CA 94720, USA. [6] School of Physics, National Key Laboratory of Crystal Materials, Shandong University, 27 Shanda South road, Jinan 250100, China. [7] Energy Lab, Samsung Advanced Institute of Technology, 130, Samsung-ro, Yeongtong-gu Suwon-si, Gyeonggi-do 16678, South Korea. [8] Department of Computer Science, Stanford University, 353 Serra Mall, Stanford, CA 94305, USA. [9] Energy1lab, Samsung SDI, 130, Samsung-ro, Yeongtong-gu Suwon-si, Gyeonggi-do 16678, South Korea. [10] Stanford Institute for Materials & Energy Sciences, SLAC National Accelerator Laboratory, 2575 Sand Hill Road, Menlo Park, CA 94025, USA. Correspondence and requests for materials should be addressed to M.F.T. (email: mftoney@slac.stanford.edu) or to W.Y. (email: WLYang@lbl.gov) or to D.P. (email: dgprendergast@lbl.gov) or to W.C.C. (email: wchueh@stanford.edu)

A more efficient and sustainable energy infrastructure requires a significant improvement in the cost and energy density of Li-ion batteries[1,2]. Lithium-rich layered oxide electrodes ($Li_{1+x}M_{1-x}O_2$, $0 < x \leq 1/3$) have attracted interest as they offer access to substantially higher energy densities than conventional layered oxide electrodes[3,4] owing to the presence of a reversible anionic redox couple in addition to the usual transition metal (TM) redox couples[5–15]. Understanding the mechanism of anion redox in these systems is necessary to mitigate their unfavorable electrochemical properties[16,17], which include significant charge/discharge voltage hysteresis (several hundred millivolts) and long-term voltage fade, even when cycled at vanishingly low rates[18–20].

Establishing the oxygen redox mechanism in these electrode materials, however, has proved challenging. Tarascon and coworkers have suggested that oxidation of oxygen generally results in the pairing of O ions, resulting in an effective $2O^{2-}/O_2^{n-}$ redox couple[5,9–14,21], which is stabilized against evolution as oxygen gas in the presence of $4d$ and $5d$ TMs due to the increased TM–O hybridization and improved band alignment over $3d$ TMs[22]. Ceder and coworkers, meanwhile, have predicted that O–O dimerization should only occur in the presence of $d^0$ and $d^{10}$ cations due to the rotational freedom of the $O_{2p}$ orbitals, with a localized hole $O^{2-}/O^-$ mechanism prevailing elsewhere[23]. Bruce and coworkers recently reported evidence for the localized $O^{2-}/O^-$ mechanism in Li- and Mn- rich Ni/Mn/Co layered oxides (LMR-NMC) based on O K edge XAS[6,7], yet similar observations have led others to conclude a $2O^{2-}/O_2^{n-}$ redox couple in the same system[12,13]. Thus, a clear consensus on which mechanism prevails in which materials is lacking.

A key factor giving rise to the confusion is the difficulty of probing anionic redox species with conventional spectroscopic techniques, given the heterogeneous nature of many Li-rich materials. For example, Li-rich surface and near-surface regions often exhibit oxygen evolution and reconstruction during delithiation[24–28], yet surface-sensitive X-ray photoelectron spectroscopy (XPS) is often used to infer the nature of bulk anion redox[5,9,11,29]. Likewise, correlations between spatially averaged X-ray absorption spectra (XAS) are often used to assess hybridization[6,7,10,12–14,30–32], which assumes that redox chemistry occurs uniformly. The relatively subtle evolution in the spectra during anion redox, combined with their sensitivity to probing depth (a few nm for quantitative XPS and total-electron-yield (TEY) XAS) and detection mode (e.g., self-absorption distortions in fluorescence-yield (FY) XAS) has contributed to the conflicting proposed mechanisms for oxygen redox in certain materials and, consequently, explanations for its role in determining capacity and electrochemical stability[15,21,22,32,33].

Furthermore, these mechanisms are typically discussed without considering the evolution of local and average structures during delithiation[6,7,22,32]. Indeed, X-ray diffraction (XRD)[34–40], transmission electron microscopy (TEM)[29,37,41], neutron diffraction[42], magnetic susceptibility[37,38,41,43], and computational[44–47] studies have confirmed that the bulk structure in many Li-rich oxides evolves substantially during delithiation, primarily due to TM migration into Li sites between the TM-O layers (Supplementary Fig. 1). Tarascon and coworkers have furthermore shown that the electrochemical signature of oxygen redox strongly suggests a coupling with structural evolution[16]. Nonetheless, the mechanistic link between structure dynamics and anion redox chemistry and stability remains largely unexplored and unclear, due to the aforementioned characterization challenges. Indeed, this connection is expected to be significant, as Ceder and coworkers have recently predicted that local coordination environment plays a crucial role in determining the oxygen redox potential[23]. Thus, clarifying the nature of anion redox and its effect on electrochemical stability requires an approach that simultaneously probes the spatial distribution of anion redox chemistry and the evolution of local structure.

In this work, we address this challenge by combining scanning transmission X-ray microscopy and nanoscale XAS (STXM-XAS) with resonant inelastic X-ray scattering (RIXS) and various structural probes to investigate the anion redox mechanism in LMR-NMC. The combination of STXM-XAS and RIXS provides the nanoscale distribution and emission signature of the features observed through conventional XAS, offering unprecedented insights into the nature of the oxidized oxygen species. We reveal that LMR-NMC exhibits two distinct reversible redox mechanisms during the first charge: (1) depopulation of hybridized $TM_{3d}$–$O_{2p}$ bands whose redox potential is relatively constant ("TM–O redox") below 4.50 V; (2) depopulation of states with predominantly $O_{2p}$ character ("O redox") above 4.50 V whose redox potential is dynamic and strongly coupled to TM migration and an associated change in the electronic structure of the material. We unambiguously confirm that both occur throughout the bulk of the primary particles, while oxygen evolution occurs only at high voltage in near-surface regions. We show that O redox persists for hundreds of cycles in uncoated LMR-NMC, which runs counter to recent predictions made for $3d$ systems under the assumption of a non-evolving structure[21,22,33]. Ab initio calculations reveal that the coupling between TM migration and O redox arises due to the dramatic change in local O coordination environment, which shifts the depopulated $O_{2p}$ states to higher energy and lowers the O redox potential relative to that of the hybridized TM–O redox by > 1 V. Thus the O redox chemistry in LMR-NMC cannot be understood as a static $O^{2-} \rightarrow O^- + e^-$ redox couple, but rather as a dynamic structure-redox coupled process described by $\{O^{2-} + TM\} \rightarrow \{O^- + TM_{mig}\} + e^-$. We suggest that this previously unconsidered structure-redox coupling plays an important role in stabilizing anion redox in LMR-NMC. Our results further suggest that anion redox chemistry can be tuned through control of the crystal structure and resulting TM migration pathways, providing an alternative route to improve Li-rich materials without altering TM–O bond covalency through substitution with heavier $4d$ and $5d$ TMs. More generally, we reveal a dynamic switching in cation and anion redox potentials, demonstrating the crucial role of simultaneous local and electronic structure rearrangement in determining electrochemical functionality in nominally topotactic intercalation materials.

## Results

**First cycle structural and electrochemical behavior.** Uncoated secondary particles of $Li_{1.17}Ni_{0.21}Co_{0.08}Mn_{0.54}O_2$ were synthesized, as described in the Methods section. Rietveld refinement of the average structure in the $C2/m$ space group to the synchrotron XRD pattern of the pristine powder yielded lattice parameters and Li-TM mixing (~2%) (Supplementary Fig. 2), consistent with previous literature[35,36,42]. We note that a two-phase model consisting of a trigonal $R\bar{3}m$ phase and a monoclinic $C2/m$ phase can also be used to describe the pristine LMR-NMC structure with similar accuracy[42]. However, similar Li-TM mixing is typically observed in these two-phase refinements, and so to improve the reliability of this refined parameter we employ the single-phase model which has fewer refined variables. Implications of the intermediate range in-plane cation ordering on the migration phenomenon investigated here are discussed later. To avoid Li transport limitations, electrochemical measurements were performed at a 4 mA g$^{-1}$ current density (~C/68 where 1 C indicates the rate it takes to charge or discharge the electrode in 1 h), unless otherwise noted. Figure 1a shows the first and second cycle voltage and dQ dV$^{-1}$ profiles. The sloping region during the first

charge is entirely reversible if the material is cycled below 4.35 V, i.e., the peaks in dQ dV$^{-1}$ profile are symmetric between charge and discharge. Upon charging to 4.60 V, the LMR-NMC electrode loses the honeycomb-like in-plane TM ordering in the TM layer, which manifests as the disappearance of the 9–11° (20–23° in Cu Kα) superstructure peaks in the XRD pattern (Fig. 1b)[48]. Simultaneously, structure refinement shows a substantial increase in the fraction of TMs in the Li layer from 2.8 % to 9.0 % (Fig. 1c, also see Supplementary Table 2), consistent with previous reports (see Methods)[35,36]. Upon subsequent discharge, the TM occupancy in the Li layer (4.7 %) does not return to that of the pristine material, the superstructure is not recovered, and the voltage profile is likewise permanently lowered in subsequent cycling. We note that this intra-cycle partially reversible bulk TM migration, which has been observed previously[34–36,40,43], is distinct from the

permanent TM migration that occurs at the surface during the first cycle, which is known to be the result of oxygen evolution and densification[27,49–51], and from the longer-term irreversible TM migration associated with the formation of a spinel-like structure and voltage fade[52]. Reversible TM migration is consistent with the TM site preference varying with electron count and/or Li content[53], such that on charge certain electronic and structural conditions are met that promote TM migration, while on discharge these conditions are quenched by lithiation and the original TM sites become favored again. The partial irreversibility (i.e., the resulting in-plane disorder and residual TM occupancy in the Li layer) suggests some "trapping" of TM ions as well as hysteresis in the migration pathways during charge and discharge.

**Oxygen oxidation during first charge voltage plateau.** To understand the unusual redox chemistry of LMR-NMC we first investigate gross changes in the electrode electronic structure by following the average spectral evolution during the first cycle. X-ray transparent samples were prepared ex situ through sonication and dispersion of electrodes harvested at various voltages (Fig. 2a) in an Ar glovebox and transported to the microscope with minimal air exposure. The spatially averaged transmission XAS spectra obtained through STXM (Fig. 2c–f), which represent the true depth-averaged absorption spectra, show that when charging below the plateau (pristine, 'P' → 4.35 V, '1') we observe primarily an inversion in the Ni L$_3$ edge peak ratio and a shift of the Co L$_3$ peak by + 0.4 eV, correlated with the simultaneous growth of a peak at 528.5 eV in the O K pre-edge. These changes, supported by the TM K edge XAS (Supplementary Fig. 3), are well-understood to reflect the depopulation of hybridized O$_{2p}$–TM$_{3d}$ (antibonding) bands[54–57], constituting standard hybridized "TM–O redox".

By contrast, when charging through the voltage plateau (4.35 V, '1' → 4.60 V, '2'), the TM L$_3$-edge spectra change minimally: the Co and Mn L$_3$ edges exhibit some broadening, and there is a small increase of the low energy peak at the Ni L$_3$ edge, suggesting a surprising reduction of the Ni during charge[12,30,31]. We later return to this important observation. By far the most significant change during the voltage plateau is the growth of a sharp and intense peak at 530.8 eV in the O K-edge spectrum, which indicates that predominantly O$_{2p}$ character states are depopulated during the 4.50 V plateau[56,58,59]. The differential spectra in Fig. 2c clearly show that this feature is distinct from the peak at 528.5 eV, confirming that the two redox mechanisms are fundamentally different. We therefore refer to the 530.8 eV peak being the signature of "O redox", the nature of which we now investigate.

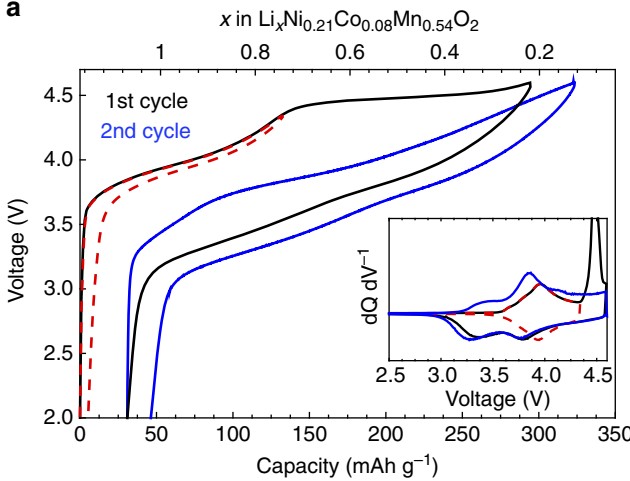

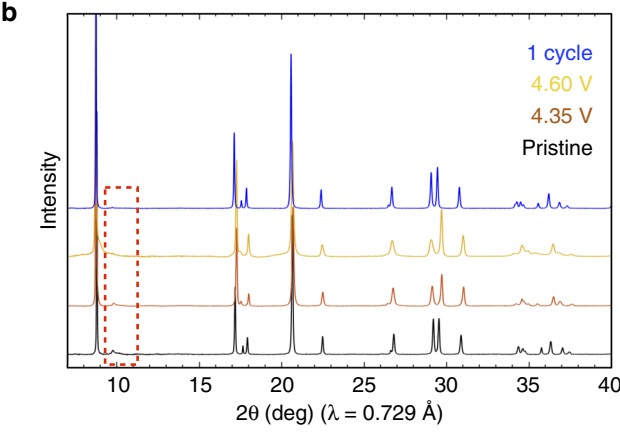

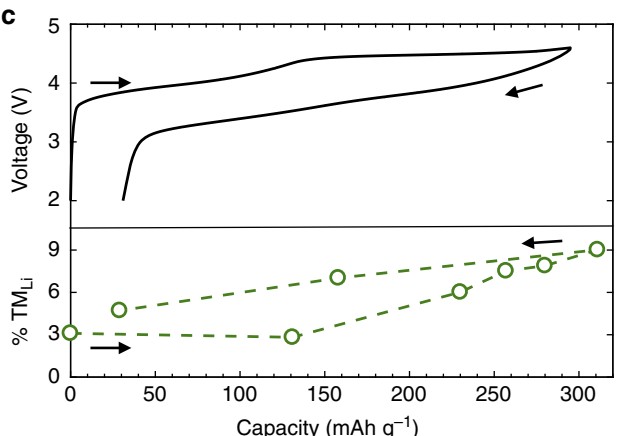

**Fig. 1** First cycle electrochemical and structural behavior of LMR-NMC. **a** First and second cycle voltage profiles of LMR-NMC at a 4 mA g$^{-1}$ current density, showing the irreversible drop in voltage after charging through the 4.50 V plateau. The red dashed trace is the voltage profile when the material is cycled below 4.35 V. Inset: dQ dV$^{-1}$ plots. Voltages here and in all other plots are measured against a lithium counter electrode. **b** Synchrotron XRD patterns throughout the first cycle showing the loss of in-plane superstructure peaks (red dashed box) after charging through the 4.50 V plateau. The in-plane structure is not recovered during discharge, indicating an irreversible rearrangement in the cation ordering. **c** Fraction of the TMs in the Li layer (TM$_{Li}$) throughout the first cycle, obtained through refinement of the crystal structure to the XRD patterns, showing the substantial increase in out-of-plane disorder during the plateau. The TM migration is hysteretic and the material never fully recovers the pristine structure. Error bars representing the refinement error are smaller than the data symbols and are therefore omitted

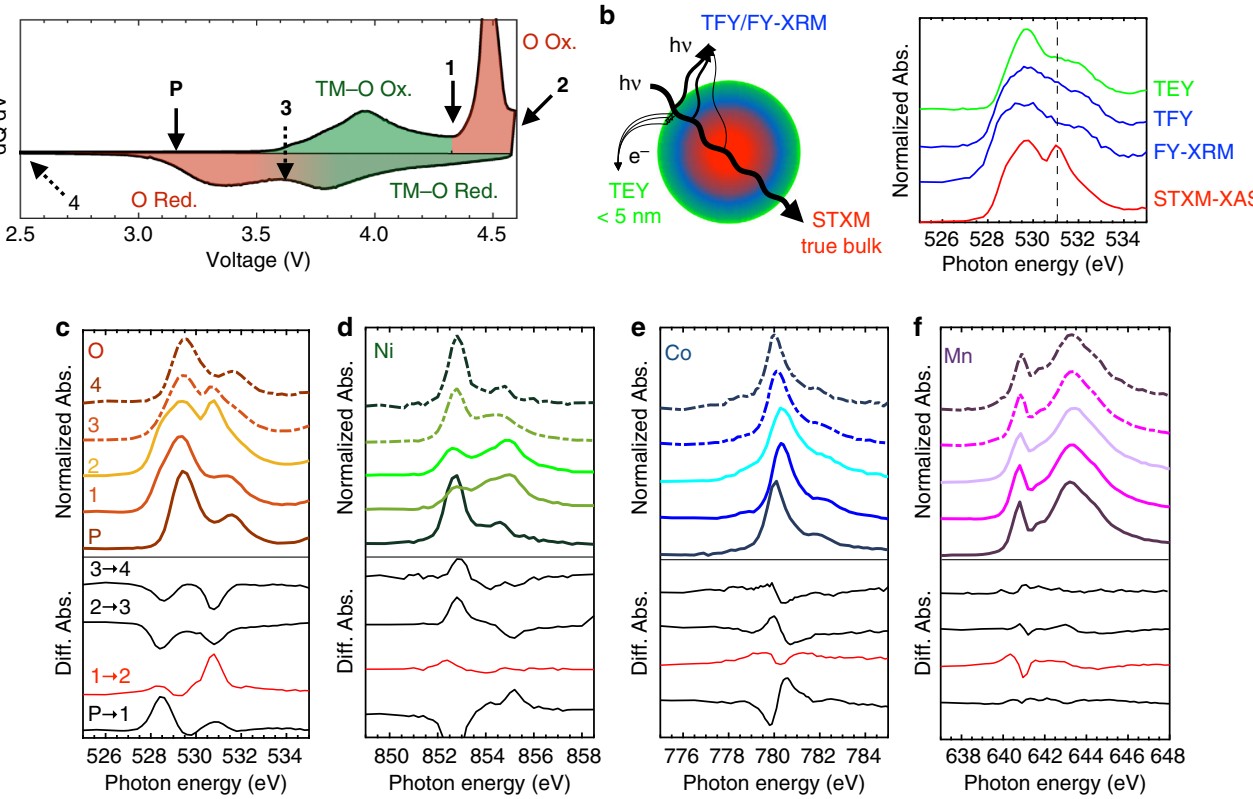

**Fig. 2** First cycle average STXM-XAS of LMR-NMC. **a** dQ dV$^{-1}$ of the first cycle showing the voltages at which samples were harvested for STXM. The samples are pristine (**P**), 4.35 V (**1**), 4.60 V (**2**), 3.65 V (**3**), and 2.00 V (**4**). Regions of the dQ dV$^{-1}$ are shaded to show the hysteresis in the O redox relative to the TM–O redox. **b** Schematic and spectra comparing STXM-XAS to various other XAS detection modes (TEY, FY, and fluorescence yield X-ray microscopy, FY-XRM[87]). The FY-XRM spectrum was collected on the same sample as the STXM spectrum, confirming sample preparation is not the cause of the discrepancy. **c–f** Spatially averaged transmission soft XAS throughout the first cycle at the **c** O K, **d** Ni L$_3$, **e** Co L$_3$, and **f** Mn L$_3$ edges, obtained by summing the spectra of all the pixels containing active material in each STXM spectro-image. Solid traces indicate electrodes harvested during charge, while dashed traces indicate electrodes harvested during discharge. The red and black traces below each plot indicate the differential spectra between the points indicated

We note that this is the first report of such a significant new peak emerging at the O K edge during the voltage plateau. Previous XAS results[6,7,10,12–14,30–32] mainly utilized electron and fluorescence based detection modes (Fig. 2b), which are distorted by surface contributions, self-absorption, and peak broadening effects and show only a weak and broad feature at the same energy. As a result, a definitive assessment of the plateau redox mechanism has been difficult. Figure 2b shows that we can reproduce these previously reported FY- and TEY-XAS results on the same electrodes, confirming that the discrepancy arises from the detection modes employed. By directly measuring the absorption coefficient, transmission XAS yields the true absorption spectrum.

**O redox is a bulk phenomenon.** To understand the chemical nature of O redox, it is important to know whether it is a bulk or surface phenomenon and whether it is spatially correlated to the subtle changes in the average TM spectra, which could be occurring in different regions of the electrode particles. We therefore map the nanoscale distribution of the Ni L$_3$- and O K-edge spectroscopy at the single primary particle length scale during the first-charge voltage plateau. In Fig. 3 we apply principal component analysis (PCA) and non-negative matrix factorization (NMF) to a series of STXM spectro-images taken at six intermediate points along the 4.50 V plateau to identify the end-member spectra that describe the spectroscopic changes. At both the O and Ni edges, two end-member spectra were identified

(Fig. 3a): the O spectra are essentially with and without the peak at 530.8 eV, while the Ni spectra resemble Ni$^{4+}$ and Ni$^{2+}$[54]. Using these end-member spectra, we obtain the average fraction (Fig. 3b) and nanoscale map (Fig. 3c) for primary particles throughout the voltage plateau. Maps of the end-members at the Mn and Co L$_3$-edges reveal only minor variations during the plateau (Supplementary Fig. 4).

Although all pixels measured with STXM do contain some surface signal, the surface contribution decreases with particle thickness, being lowest in the center of the particles' 2D projections. Since the ellipsoidal LMR-NMC particles are several hundreds of nanometers in thickness in the center, contribution from the surface (a few nanometers) is small, and the signal is dominated by the bulk. For the same reason, near the edge of the particles, surface signal dominates. Thus, from the nanoscale maps of the O end-member spectra (Fig. 3c, top), it is immediately evident that O oxidation occurs throughout the bulk of the primary particles during the voltage plateau, establishing unambiguously that O redox is a bulk process in LMR-NMC. On the other hand, the Ni chemical maps (Fig. 3c, bottom) reveal that bulk Ni ions remain in a 4 + oxidation state during the plateau. The variation observed in the average spectrum in Fig. 2d is in fact due to Ni reduction that is confined mostly to the primary particle surfaces. Spectral linescans in Fig. 3c show that the new peak at 530.8 eV at the O K edge is actually suppressed near the particle surfaces where Ni is reduced. In other words, although Ni reduction and O oxidation occur simultaneously during the voltage plateau,

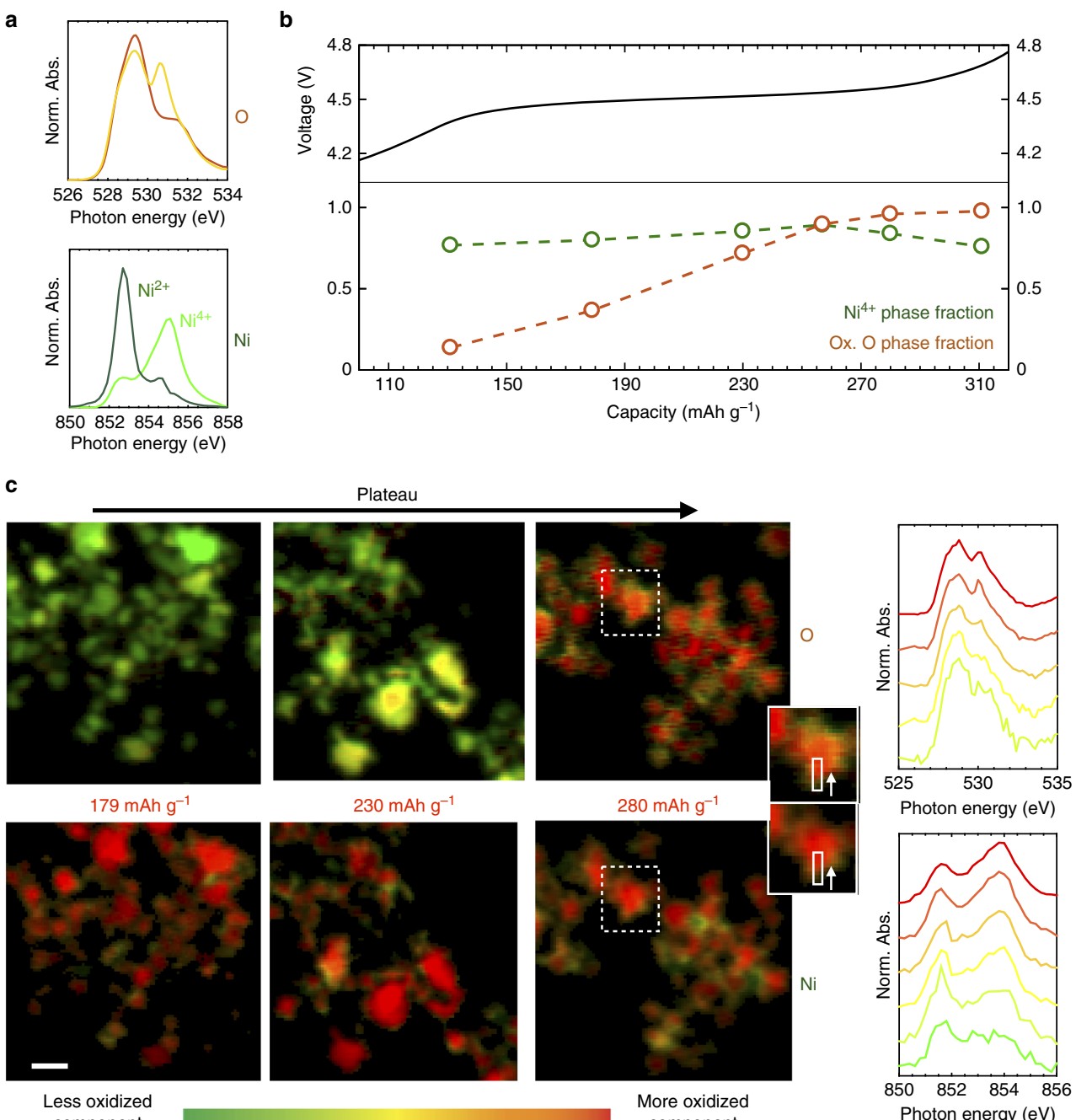

**Fig. 3** Spatial dependence of O and Ni spectroscopic response during 4.50 V plateau. **a** End-member spectra obtained from NMF on the STXM spectro-images at the Ni L$_3$ and O K edges throughout the first cycle plateau. **b** Total end-member fractions for each sample used in the NMF analysis as a function of capacity through the plateau. **c** Nanoscale distribution of the end-members in **a** for primary particles at different points in the voltage plateau. Scale bar is 500 nm. Right: spectral line-scans of the magnified particle showing the bulk O oxidation and surface Ni and O reduction. The spectra at the bottom correspond to the near-surface region and those at the top correspond to the bulk

they are spatially separated. Its prevalence near the surface suggests that Ni reduction is more likely due to a steep gradient of oxygen non-stoichiometry due to oxygen evolution and material densification or reaction with the electrolyte[6,7,25,27,50,60]. This is further confirmed by the redox behavior during the first discharge and second charge (Fig. 4a), which reveals that discharging to 2.00 V reversibly reduces Mn near the surface with no change in the bulk. This is consistent with an electrochemically active but oxygen deficient near-surface structure that approaches the bulk composition within tens of nanometer.

**O redox is non-rigid and stable in LMR-NMC.** The spatially resolved nano-spectroscopy afforded by STXM reveals minimal pixel-wise correlation between the TM and O spectroscopy in the bulk during the voltage plateau, seen quantitatively in Fig. 4b and also in Supplementary Fig. 5. Thus, changes in the average TM spectroscopy do not offer insight into the nature of O redox, as previously proposed[12]. However, the nature of the O redox XAS signature can provide such insight, as changes in the O K edge can reflect how the O$_{2p}$–character states are altered during the redox process[58,61–63]. For example, Mueller et al.[56] concluded from O K edge XAS that O redox in Fe-containing perovskites

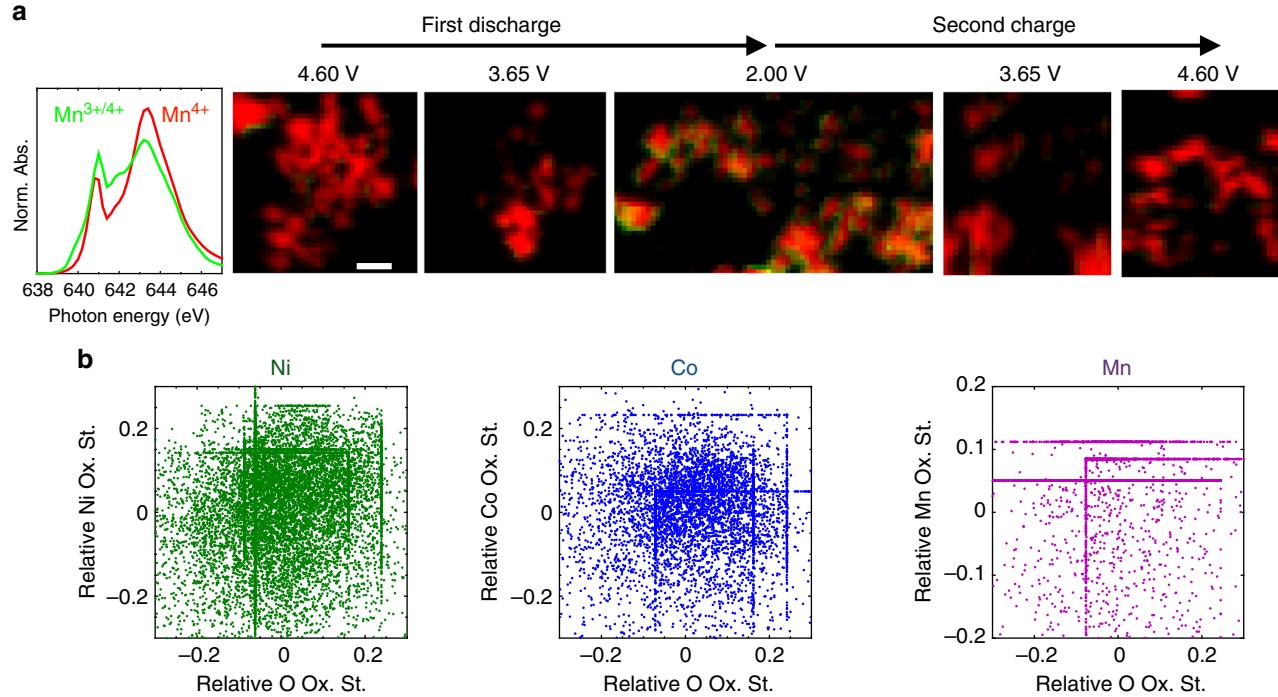

**Fig. 4** Spatial separation of O oxidation and TM reduction. **a** Maps of the two Mn end-member spectra (left) obtained by NMF throughout the first discharge and second charge, showing the reversible reduction of Mn only at the particle surfaces upon discharge to 2.00 V. The Mn oxidation states are assigned based on previous literature[88]. Scale bar is 500 nm. **b** Scatter plots of the single-pixel O oxidation state (i.e. fraction of the fully oxidized O phase) vs. the TM oxidation state (i.e., fraction of $TM^{4+}$) throughout the first charge voltage plateau, showing minimal correlation between the O and TM oxidation states during O oxidation. The single-pixel TM and O oxidation states for each sample along the plateau are plotted relative to their respective means for each sample to generate the plots. The means for each sample at each edge can be seen in Fig. 2b and Supplementary Fig. 4a. The straight lines are due to pixels with 100% or 0% of a given oxidation state, shifted by the mean of the sample to which those pixels correspond

occurs via reversible depopulation of a narrow, rigid, mostly O-character $TM_{3d}$–$O_{2p}$ band near the Fermi level. Indeed a similar mechanism has been proposed for O redox in LMR-NMC, constituting an $O^{2-}/O^{-}$ redox couple in the limit where the $O_{2p}$ character approaches 100%[6,7]. However, unlike the perovskite system, the O redox peak observed here (530.8 eV) does not appear at the absorption onset (~528 eV) but above it. In fact, the O redox state appears above the depopulated TM–O redox state (528.5 eV), suggesting a switch in their relative positioning after O oxidation. Thus, O redox in LMR-NMC is linked to a relative reordering of the anion and cation electronic states that cannot be described by a static $O^{2-}/O^{-}$ redox couple.

The significant change in the $O_{2p}$ states after depopulation is further confirmed in Fig. 5a, which plots the O K-edge RIXS obtained throughout the first cycle. RIXS maps the fluorescence intensity as a function of both absorption and emission energy[64], revealing the energy distribution buried in the features observed through STXM-XAS. Rapid acquisition of such extensive RIXS maps was only made possible by the recent commission of ultra-high efficiency RIXS systems[65]. The Supplementary Information includes a detailed discussion of the RIXS features, with the relevant details summarized here. In the pristine electrode, excitation into the unoccupied hybridized $TM_{3d}$–$O_{2p}^{*}$ (528–533 eV) and $TM_{4sp}$–$O_{2p}^{*}$ (> 535 eV) states results in similar emission in the 522–527 eV range, corresponding to a decay from the relatively broad (delocalized) oxygen valence band states to fill the excited $O_{1s}$ core hole, as extensively observed and discussed in other TM oxides[66]. The RIXS map of the electrode at 4.35 V shows that emission at 528.5 eV excitation energy from states depopulated during TM-O redox is identical to that for the existing unoccupied $TM_{3d}$–$O_{2p}^{*}$ states in the pre-edge, confirming that they are chemically similar. For the electrode at 4.60 V,

however, excitation to the new unoccupied O redox state at 530.8 eV results in a striking new emission feature at 523.25 eV, which is clearly distinct from the broad 522–527 eV emission features from the TM–O hybridized states. Thus, RIXS is able to clearly differentiate the chemical nature of these states by providing the emission signature of the key 530.8 eV peak observed through STXM-XAS. The distinct RIXS feature created during the 4.50 V plateau further supports the scenario of reshuffled electron states after O redox, which cannot be explained by a rigid $O^{2-}/O^{-}$ mechanism.

Furthermore, such a mechanism was previously predicted to be unstable against oxygen evolution in 3d layered oxides[21,33]. We therefore investigate the stability of oxygen redox over extended cycling using optimized 18650 cells (Samsung, see Methods), which exhibit 94% retention of their low-rate (intrinsic) capacity after 500 cycles at a 1C/2C charge/discharge rate. Figure 5b plots the O RIXS maps for the 501st cycle in the charged and discharged states and shows that even after 500 cycles the oxygen redox feature still reversibly appears. Thus, a significant fraction of the oxygen anions remains redox active after hundreds of cycles. These particles were not protected against oxygen evolution through coating. This is the first spectroscopic evidence of long-term stability of anion redox in 3d Li-rich materials and further suggests that the unstable rigid $O^{2-}/O^{-}$ mechanism is unlikely.

**Electronic reshuffling inverts redox sequence on discharge**. The dynamic reshuffling of the O redox states has a profound effect on the LMR-NMC electrochemistry. Figure 2 reveals that the spectra on discharge (4.60 V, '2' → 3.65 V, '3' → 2.00 V, '4') do not follow a simple reversal of the changes observed during charge.

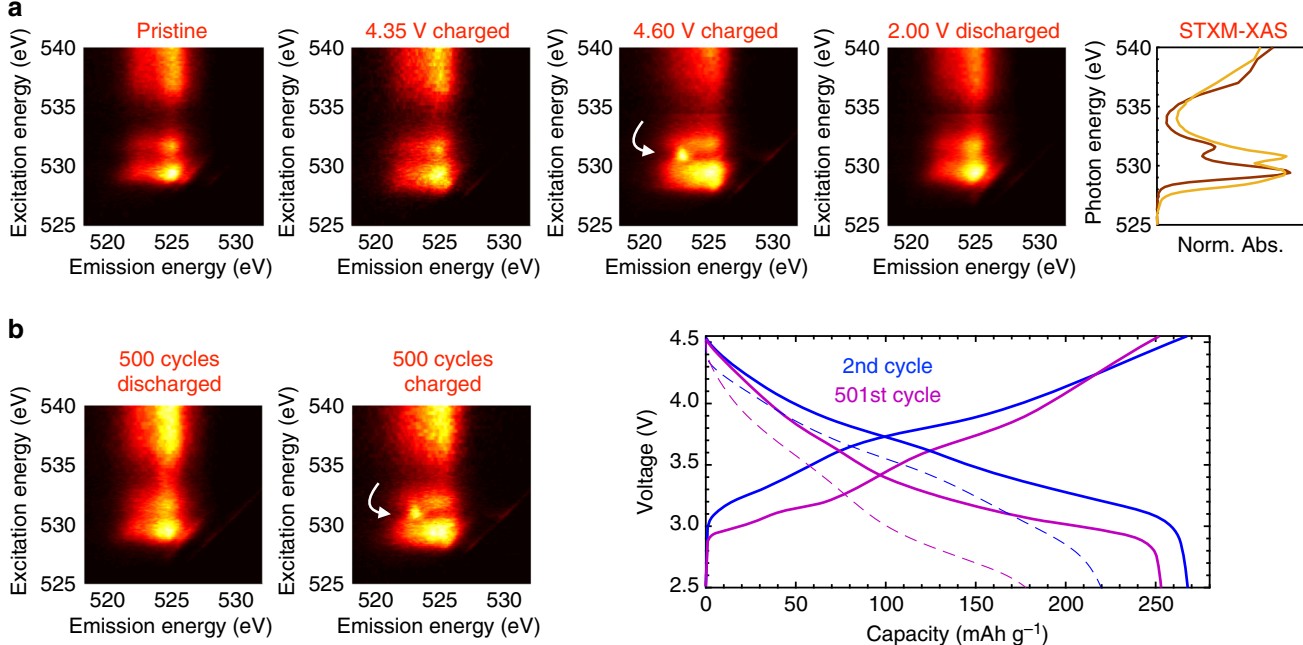

**Fig. 5** Probing the nature and stability of oxygen redox. **a** RIXS maps at the voltages indicated throughout the first cycle. The unique emission signature at 4.60 V indicated by the white arrow supports an electronic restructuring associated with O redox. Right: XAS obtained in the pristine (brown) and fully charged (tan) state during the first cycle for comparison. **b** RIXS maps acquired before and after the 501st charge on an electrode cycled 500 times at 1C/2C charge/discharge rate, showing that the reversible oxygen redox feature persists for hundreds of cycles. The voltage curves for the second and 501st cycles at C/68 (solid) and 2C (dashed) show that most of the capacity fade over 500 cycles is from increased impedance and overpotential, and that the intrinsic capacity is largely retained

Most notably, whereas O oxidation took place exclusively above 4.35 V during charge (after the Ni and Co had been fully oxidized to ~ 4 + ), during discharge the Ni and Co are almost entirely reduced first in the 4.60–3.65 V range ('2 → '3'), with most of the O reduction taking place later in the 3.65 − 2.00 V range ('3' →'4'). This massive > 1 V shift in the O redox voltage is illustrated by the color-coded regions in the dQ dV$^{-1}$ plot in Fig. 2a. Thus the sequence of the redox couples (i.e., the order in which electronic states are (de)populated) is inverted after the 4.50 V plateau, and the inverted sequence (relative to the first charge) persists into the subsequent cycles (Supplementary Fig. 6), consistent with the altered structure and electrochemistry after the first cycle (Fig. 1). Spectro-imaging confirms that this is a bulk phenomenon (Supplementary Fig. 7). This agrees well with our conclusion from XAS that some of the O$_{2p}$ states depopulated during the plateau are shifted from an energy below the TM$_{3d}$–O$_{2p}$ antibonding states to one above them, further confirming the electronic reshuffling associated with O redox.

**O redox voltage modulated by TM migration.** Figure 6a reveals a clear correlation between the hysteresis in the O redox and the fraction of TMs in the Li layer as calculated from Rietveld refinement of the synchrotron XRD patterns, suggesting that the electron state reshuffling arises from a strong coupling between O redox and TM migration. Supplementary Fig. 8 further shows that this structure-redox link is only present for the O redox and not the TM–O redox. To understand this coupling, we perform density functional theory calculations using HSE06 functionals on a Li$_{28}$TM$_{20}$O$_{48}$ (i.e. Li$_{1.17}$TM$_{0.83}$O$_2$ where TM = Mn, Ni) model supercell (Fig. 6b) to explore the effect of TM migration on the O projected density of states (pDOS). We did not include Co in our calculations due to its low concentration in our experimentally studied material. Ceder and coworkers have recently shown that

anion redox chemistry is most strongly affected by the anion nearest-neighbor coordination environment[23]. We therefore designed the supercells to contain both possible oxygen coordination environments in LMR-NMC: O$^{(1)}$, coordinated to two TMs and exhibiting the Li–O–Li geometry[23], and O$^{(2)}$, coordinated to three TMs. While there is a debate as to the intermediate range clustering of the TMs in LMR-NMC (i.e., whether it is a nano-composite of monoclinic Li$_2$MnO$_3$ and rhombohedral NMC domains[3], or a uniform solid solution monoclinic structure[67]), we note that in both cases there are only the two types of oxygen environment mentioned, and thus the effect of TM migration on anion redox in either case can be understood by observing the effect on O$^{(1)}$ and O$^{(2)}$ sites. We first show in Fig. 6b that deep delithiation (to Li$_0$TM$_{20}$O$_{48}$ or Li$_0$TM$_{0.83}$O$_2$) depopulates specific high-energy O$_{2p}$ states on the two TM-coordinate O$^{(1)}$ sites. The integrated charge density of these unoccupied states resembles pure oxygen 2p orbitals lying along the Li–O–Li axes, as first demonstrated by Ceder and coworkers[23]. In the absence of further structural change, this represents the static O$^{2-}$/O$^-$ model. As we have shown, this alone cannot explain the dynamic spectroscopic or electrochemical behavior of LMR-NMC.

Figure 6b also shows the DOS projected onto the two different oxygen ions in this delithiated state after migration of either Ni or Mn, ending with the TM in an octahedral site in the Li layer. Similar migration pathways have been shown to have reasonable migration barriers (as low as 0.1 eV) in Li$_2$MnO$_3$[68,69]. The effect of other migration scenarios is shown in Supplementary Fig. 9. Detailed discussion of all pathways investigated in this study is included in Supplementary Methods. While our treatment here is by no means exhaustive, we find that TM migration in all cases results in a shift to higher energy of the O pDOS regardless of the pathway examined and the identity of the migrating TM, suggesting that this observation is general and largely independent of the migration mechanism. We propose that this is due to

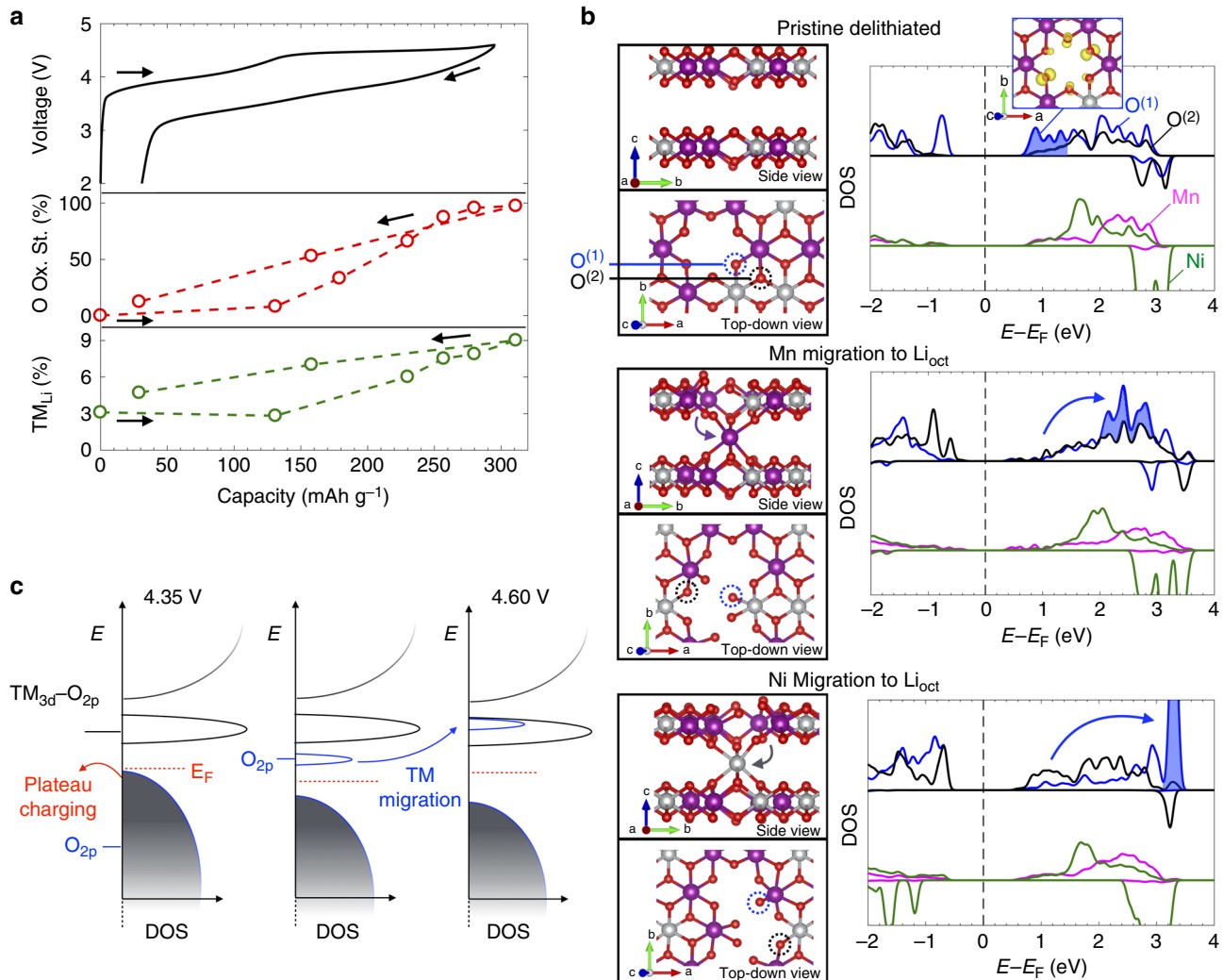

**Fig. 6** Effect of TM migration on electronic structure. **a** Plot of the O fractional oxidation state (red) and the migrated TM fraction (green) as a function of capacity, showing the clear link between hysteresis in the TM migration and voltage hysteresis in the O redox. Error bars indicating fitting residual and refinement error for the O oxidation state and TM migration fraction, respectively, are smaller than the data symbols and are therefore not shown. **b** pDOS for the TMs and the two-coordinate ($O^{(1)}$, blue) and three-coordinate ($O^{(2)}$, black) oxygen environments in the pristine delithiated state (top), and after Mn (middle) and Ni (bottom) migration into octahedral sites in the Li layer. The integrated charge density for the lowest unoccupied states in the pristine delithiated structure (blue shaded area of the pristine DOS) is shown in the top right inset. The total TM pDOS is normalized by the number of TMs in the supercell for comparison with the single-site oxygen pDOS. Schematics of each supercell used to generate the pDOS are shown to the left, with the plotted $O^{(1)}$ and $O^{(2)}$ oxygen environments circled. **c** Schematic of the reorganization of the electronic structure due to TM migration

the significant change in electrostatic environment, which necessarily modulates the oxygen redox potential as Chen and Islam have shown for the surface of $Li_2MnO_3$[44]. An oxidized oxygen that was initially bonded to two TMs ($O^{(1)}$ in Fig. 6b) can become singly coordinated, with its depopulated $O_{2p}$ states raised by several eV into the TM redox band. An oxygen that was initially unoxidized and bonded to three TMs ($O^{(2)}$) can become doubly coordinated and oxidized, transferring its electrons to another O or a TM. K-edge XAS (Supplementary Fig. 3, cf. ref. [40]) and extended X-ray absorption fine structure (EXAFS) measurements (Supplementary Fig. 10) indicate that Ni migrates the most during the voltage plateau, suggesting an important structural role of Ni in controlling the O redox chemistry. Indeed, the shift of the $O_{2p}$ states appears to be greater when the Ni migrates and the $O^{(1)}$ remains bonded to a Mn ion. The altered coordination environment of the shifted $O_{2p}$ states due to TM migration is consistent with the distinct RIXS signature, while the shifting to higher energy is consistent with the inverted bulk redox sequence

observed by STXM-XAS as well as the strong correlation between oxygen oxidation state and TM migration observed through XRD.

While we only observe a relatively small increase of TM in the Li layer (3–9%) during the plateau, each migration can de-coordinate up to five oxygen ions (assuming the TMs move into neighboring octahedral sites), meaning that up to ~20% of the oxygen sites can be directly affected. We also observe smaller indirect shifts in the oxygen redox states over longer ranges due to the distortion of the oxygen sub-lattice as a result of migration, and thus an even greater fraction of oxygen sites can be affected. We also note that in-plane TM rearrangement, which we observe as the loss of the superstructure peaks during the first cycle in Fig. 1b but cannot quantify due to their weak intensity and asymmetric shape, can have the same effect of reducing the overall O coordination number if, for example, the TMs move into tetrahedral sites. Thus additional in-plane TM migration beyond the 3–9 % out-of-plane migration may contribute to shifting a larger fraction of the O redox to lower voltage.

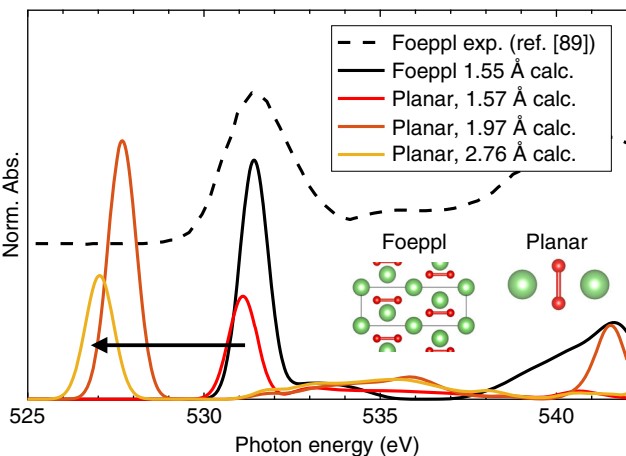

**Fig. 7** Ab initio XAS of $Li_2O_2$ as a function of O–O separation. Experimental and calculated XAS of a relaxed $Li_2O_2$ crystal in the Foeppl structure[89] (middle) with an O–O bond length of 1.55 Å, and calculated spectra of an artificial planar $Li_2O_2$ molecule (right) with various bond lengths. The molecule was first allowed to relax, yielding the 1.57 Å O–O bond length, which was then stretched artificially to create the subsequent molecular structures. Ab initio XAS was performed in the spin polarized DFT-XCH framework, as described previously[90]

## Discussion

We therefore conclude that TM migration is intrinsically coupled to the bulk O redox potential, necessarily causing the reshuffling of the relative O and TM–O redox potentials. Thus, the O redox mechanism in LMR-NMC cannot be described as a static $O^{2-}/O^-$ redox couple, but rather as a dynamic $\{O^{2-} + TM\} \rightarrow \{O^- + TM_{mig}\} + e^-$ process, where $TM_{mig}$ indicates a migrated TM. In the specific case where a nominally 4 + Ni ion migrates into an octahedral Li vacancy in the Li layer, this can be written using Kröger-Vink notation[70] as: $Ni_{Ni}^{\times} + O_O^{\times} + V_{Li}^{'} \rightarrow Ni_{Li}^{\cdots} + V_{Ni}^{''''} + O_O^{\cdot} + e^-$. Since many different migration pathways for the various TMs are possible, we represent the substantial change and distribution in the oxygen coordination environments with $\{O^- + TM_{mig}\}$, which modulates the oxygen redox potential and shifts a large fraction to lower voltage (<3.65 V) after the first charge. Figure 6c shows a schematic representation of this redox process from an electronic structure perspective. Importantly, if the O redox voltage is a primary factor affecting its stability, as has been suggested[22], then the shifting of a substantial fraction of O redox to lower voltage coupled with TM migration may stabilize LMR-NMC against oxygen evolution, offering a possible explanation for the long-term anion redox stability observed here. Thus, while the long-term, irreversible TM migration associated with the formation of spinel-like structures is known to be associated with voltage fade[29], the intra-cycle, partially reversible bulk TM migration may in fact be beneficial for the long-term cycling of LMR-NMC. While the role of the structure-redox coupling in voltage fade requires further study, we tentatively suggest that strategies aimed at increasing the latter while minimizing the former (e.g., by screening cation substitutions) may be effective in improving the cyclability of LMR-NMC. Further work will seek to understand the relative kinetics of O redox and TM migration, identify the intermediate states, and establish whether, for example, $O_{2p}$ states are first depopulated, promoting TM migration through (de)stabilization of specific structural configurations, as has been predicted computationally[69].

We note that contraction of the O–O distance to form $O_2^{n-}$ species has also been proposed as a non-rigid redox

mechanism in certain Li-rich materials[5,9–14,21,22]. Indeed, similar XAS features at 530.8 eV have been previously assigned to such species due to the comparable energy of the $1s \rightarrow \sigma^*$ transition of $Li_2O_2$[12–14,71]. However, this transition in $Li_2O_2$ is typically broad[71], contrasting with the sharp resonance observed in LMR-NMC, and we furthermore show in Fig. 7 that the calculated XAS signature for O–O dimers only resides at this energy when they have bond lengths similar to that of $Li_2O_2$ (~1.55 Å). At the larger O–O separations that we observe in our own calculations (~1.9 Å) and at the even larger separations observed in other Li-rich materials (≥2.2 Å[5,9]) the $1s \rightarrow \sigma^*$ transition lies at much lower energy (<529 eV) and therefore cannot explain our spectroscopic observations. Furthermore, Raman[6,7] and computational studies[23] have shown the formation of O–O dimers with true ~1.55 Å bond lengths to be unlikely in LMR-NMC.

In summary, we have comprehensively established the bulk anion redox mechanism in LMR-NMC. By applying STXM-XAS and RIXS to resolve the spatial distribution and emission signatures of conventional XAS features, we overcome the long-established challenges of material heterogeneity and spectroscopic ambiguity to demonstrate the first robust quantitative detection and analysis of bulk anion redox in Li-rich layered oxides. We reveal the inherent unreliability of quantifying anion redox through measurement of the overall O K pre-edge intensity alone[6,7], which we show to be affected by both TM–O and O redox and has furthermore long been known to be heavily influenced by changes in TM–O hybridization strength[72,73]. We reveal that, unlike the lower voltage TM–O redox, the bulk 4.50 V plateau redox mechanism cannot be understood as a static redox couple. We show unambiguously that in this regime a reshuffling of the electron states and redox potentials occurs after depopulation of predominantly $O_{2p}$ character states in LMR-NMC, giving rise to unique spectroscopic signatures that are distinct from previously reported rigid-band anion redox[56]. By correlating these observations with average and local structure probes, and in conjunction with ab initio calculations, we show that this dynamic character arises from a coupling between the $O^{2-}/O^-$ redox partner and TM migration, which reorganizes the electronic structure and shifts the $O_{2p}$ states to higher energy relative to the TM–O states. This modulates the anion redox potential, inverting the TM–O/O redox sequence after the first charge and coupling the oxygen oxidation state to the fraction of migrated TMs. We therefore rewrite the mechanism as $\{O^{2-} + TM\} \rightarrow \{O^- + TM_{mig}\} + e^-$, which holistically describes the spectroscopic, electrochemical, and structural properties of LMR-NMC. We propose that this structure-redox coupling is involved in stabilizing the oxygen redox couple, which we observe to persist for 500 cycles in uncoated LMR-NMC despite predictions of intrinsic instability under the assumption of static structure. We therefore reveal that anion redox chemistry in many Li-rich materials cannot be fully understood by study of only their as-synthesized structures. Likewise, variations in structural behavior must be considered when comparing anion redox chemistry and stability between materials. Our results suggest that it may be possible to tune the stability and voltage of anion redox through control of the TM migration pathways. Thus, we suggest a new strategy for designing Li-rich layered oxides with improved cycling performance whereby the oxygen redox chemistry is tuned through structural modifications rather than the more common covalency modifications, which typically require substitution with rare 4d and 5d elements. More broadly, we demonstrate that atomic and electronic structural evolution during (de)intercalation need to be considered when assessing anion and cation redox chemistry even in nominally topotactic intercalation electrodes.

## Methods

**Materials**. The $(Ni_{0.21}Co_{0.08}Mn_{0.54})(OH)_2$ precursor powder was synthesized via co-precipitation in a continuously stirred tank reactor. Appropriate amounts of $NiSO_4$, $CoSO_4$, and $MnSO_4$ (Ni:Co:Mn = 0.21:0.08:0.54) were dissolved in deionized water with 0.2 M $NH_4OH$ as a chelating agent. Aqueous 5 M NaOH was then slowly added to the solution, precipitating the mixed hydroxide. The pH of the solution was kept greater than 10, but was otherwise not controlled. After filtering and drying, the $(Ni_{0.21}Co_{0.08}Mn_{0.54})(OH)_2$ precursor was mixed with the appropriate amount of $Li_2CO_3$ followed by calcination at 900 °C for 10 h to obtain pure $Li_{1.17}Ni_{0.21}Co_{0.08}Mn_{0.54}O_2$ powder. The Li/Ni/Co/Mn molar ratios were 1.434: 0.251: 0.099: 0.650, as measured by inductively coupled plasma mass spectrometry.

**Electrochemical measurements**. For all electrochemistry figures in the main text except Fig. 5b, 80 wt% LMR-NMC, 10 wt% polyvinylidene fluoride binder (MTI Corporation) and 10 wt% carbon black (Timical C65) were mixed with N-methyl-2-pyrrolidone (Acros Organics) and cast onto Al foil using a doctor blade with a nominal thickness of 300 μm. The film was dried at 60 °C in air for ~ 3 h followed by ~ 12 h at 100 °C under vacuum. Coin cells (size CR2016, MTI Corporation) containing a ~9 mm diameter NCM electrode, two 25 μm thick Celgard separators, a 750 μm thick Li foil anode (Sigma-Aldrich), and 1 M $LiPF_6$ in 1:1 (wt/wt) ethylene carbonate (EC)/diethyl carbonate (DEC) electrolyte (Selectilyte LP 40, BASF) were assembled. The voltage curves at 4 mA $g^{-1}$ were then measured in order to minimize the contribution of any kinetic overpotential to the measured voltage. All electrochemical measurements were performed in a temperature controlled chamber (Espec) set to 30 °C. Details on the electrochemical measurements in Fig. 5b and general ex situ sample preparation are given in the Supplementary Methods.

**X-ray diffraction**. X-ray diffraction was performed at beamline 7–2 at the Stanford Synchrotron Radiation Lightsource (SSRL) at 14 keV (0.8856 Å) beam energy. The size of the X-ray beam was $100 \times 100$ μm$^2$, and the distance between sample and the Pilatus-100K detector was 1000 mm. Electrodes were transferred to a Kapton adhesive film to remove the aluminum current collector. A pellet of the pristine powder was also measured, which showed negligible difference compared to the Kapton film sample. All of the samples were measured using the theta-2theta geometry. Rietveld Refinement details are given in the Supplementary Methods.

**X-ray absorption measurements**. XAS measurements in the hard X-ray regime were performed in transmission mode on the intact harvested electrodes (sealed under argon in a polymer pouch) at beam line 4-1 at the Stanford Synchrotron Radiation Lightsource at SLAC National Laboratory using a Si (220) double crystal monochromator detuned to 50–60% of its original intensity to eliminate high order harmonics. The spectra of Ni, Co, and Mn reference foils were used to calibrate the photon energy by setting the first crossing of the second derivative of the absorbance spectrum to be 8333 eV, 7709 eV, and 6539 eV, respectively. Three ion chambers were used in series to simultaneously measure $I_0$, $I_{sample}$, and $I_{ref}$. Spectrum normalization and alignment was performed using the Athena software package[74].

**Scanning transmission X-ray microscopy**. Harvested electrodes were sonicated in DMC under argon for 2 h at 45 °C to dislodge the LMR-NMC material from the electrode and break up the secondary particle structure. TEY XAS was performed on sonicated and un-sonicated materials and confirms that the procedure does not damage or alter the material, even in the ~2 nm near the surface probed by TEY XAS (Supplementary Fig. 11). The primary particle suspension was drop-cast onto copper TEM grids with a carbon film (Ted Pella). The grids were then loaded onto a sample holder which was sealed under argon and brought to the beam line. STXM was performed at beam line 11.0.2 at the Advanced Light Source (ALS) in Lawrence Berkeley National Laboratory[75], with preliminary measurements conducted at beam line 5.3.2.2[76]. Imaging was performed with either a 25 nm or 50 nm zone plate. To acquire an image, an interferometer-controlled stage raster-scans the position of the sample while a point detector measures the transmitted beam intensity[77]. While the step size was varied (35–80 nm) between edges and samples, the dwell time for each pixel was typically 1 ms. We confirmed that the ex situ imaging experiment did not damage the samples by measuring the Ni $L_3$ edge before and after imaging at the other three edges and observing no change to the average spectrum. STXM spectro-images were aligned in the aXis2000 software package. Subsequent data processing, including principle component analysis, was performed in the MATLAB 2016b software package and is detailed in the Supplementary Methods.

**Resonant inelastic X-ray scattering**. Soft X-ray RIXS maps (and conventional TEY and TFY XAS spectra) were collected in the newly commissioned ultra-high efficiency iRIXS endstation at Beamline 8.0.1 of the ALS[65]. All the samples were processed in glove box with high purity Ar environment, and the mounted samples were sealed and transferred into the experimental vacuum chamber through a specially designed sample transfer kit[78]. The kit was directly pumped down without stopping the pump while opening to the vacuum chamber to avoid any air exposure. Radiation damage was monitored by multiple scans of XAS and was

considered negligible for these samples. Electrode sample surfaces were mounted at 45° to the incident X-ray beam. The outgoing photon direction (momentum transfer) along the RIXS spectrograph is 90°. RIXS resolving power and other technical details of the VLS spectrograph used for this work can be find in our previous report[79]. Details on how the RIXS maps are generated are given in the Supplementary Methods.

**First-principles calculations**. Lattice relaxations were performed using the Perdew–Burke–Ernzerhof (PBE) generalized gradient approximation[80] with the rotationally invariant Dudarev method to correct for self-interaction[81] as implemented in the Vienna Ab initio Simulation Package (VASP)[82]. A U values of 3.9 and 6.0 eV were used for Mn and Ni, respectively, according to previous literature[23]. Projector-augmented wave pseudopotentials were used[83,84] with a plane-wave basis set with a kinetic energy cut-off of 600 eV. Generally, all atomic positions and lattice parameters were allowed to relax until the forces on the atoms were less than 0.01 eV Å$^{-1}$. In certain cases, lattice parameters were fixed to prevent changes in stacking order during migration that otherwise proceed with no energetic barrier (discussed below). Since the structural composition is kept constant during migration, migration enthalpies are calculated as the difference in the cell energies before and after migration. Oxygen vacancy formation enthalpies are calculated with reference to molecular oxygen gas in a $10 \text{ Å} \times 8.5 \text{ Å} \times 10 \text{ Å}$ cell. Ferromagnetic coupling was assumed.

For electronic structure calculations, spin-polarized Density Funcitonal Theory (DFT) calculations employing HSE06 functionals were performed using the Quantum Espresso software package[85,86]. An exact exchange mixing parameter of 0.25 was used for all calculations. Fritz-Harber Institute pseudopotentials were used to replace the all-electron ion potentials, and the electronic wavefunctions were described using a plane-wave basis set with a kinetic energy cutoff of 80 Ry. A k-point grid of $2 \times 2 \times 2$ was used for typical supercell dimensions of ~$10 \text{ Å} \times 8.5 \text{ Å} \times 10 \text{ Å}$. The input structures were obtained from the PBE lattice relaxations, and the nuclear positions were allowed to further relax at the HSE level, keeping the lattice parameters fixed. Details on the methodology employed are discussed in Supplementary Note 2.

**Data availability**. The data supporting the findings of this study are available from the corresponding authors on request.

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

## Acknowledgements

The research was supported by the Assistant Secretary for Energy Efficiency and Renewable Energy, Office of Vehicle Technologies, Battery Materials Research Program, U.S. Department of Energy, and by Samsung Advanced Institute of Technology Global Research Outreach program. W.E.G. was supported additionally by the Advanced Light Source Doctoral Fellowship. Q.L. thanks the China Scholarship Council (CSC) for financial support through the collaboration based on China 111 Project No. B13029. Use of the Advanced Light Source was supported by the Office of Science, Office of Basic Energy Sciences, of the U.S. Department of Energy under Contract No. DE-AC02-05CH11231. Use of the Stanford Synchrotron Radiation Lightsource, SLAC National Accelerator Laboratory, was supported by the Office of Science, Office of Basic Energy Sciences, of the U.S. Department of Energy under Contract No. DE-AC02-76SF00515. Work at the Molecular Foundry was supported by the Office of Science, Office of Basic Energy Sciences, of the U.S. Department of Energy under Contract No. DE-AC02-05CH11231. Part of this work was performed at the Stanford Nano Shared Facilities (SNSF) at Stanford University. We would like to thank M. Saiful Islam, David N. Mueller, Thomas P. Devereaux, Chunjing Jia, and Brian Moritz for insightful discussions, and Paul Kent for providing support for the computational work.

## Author contributions

W.E.G. and W.C.C. conceived the study. W.E.G. was involved in all aspects of the study other than material synthesis. K.L., K.H.S., and M.F.T. led the diffraction experiments and interpretation. Y. Liang and T.B. contributed to the computational work, which was supervised by D.P. Q.L. and W.Y. performed the RIXS measurements and W.Y. contributed to the interpretation. S.-J.A. and J.H.S. synthesized the material and cycled the 18650 cells. M.M., A.M., and S.E. contributed to the PCA and NMF analyses. J.H., Y.Li, T.T., D.K., and D.V. contributed to the STXM measurements and J.H. and Y. Li contributed to the data interpretation. W.E.G. and W.C.C. wrote the paper with significant contribution from D.P. and W.Y. Efforts at Samsung were supervised by J.-H.P. and S.-K.D. W.C.C. supervised the overall project.

## Additional information

**Competing interests:** The authors declare no competing financial interests.

