## [Peer Review File · Nature Communications]

Reviewers' comments:

Reviewer #1 (Remarks to the Author):

Overall, the experimental side of this paper is strong, providing sound evidence for oxygen redox separated from TM-O redox, clearly supporting the notion of anionic redox (without dimer formation). The most interesting point is electronic state re-arrangement occurs after oxygen oxidation redox in the bulk during the first charge voltage plateau.

1. It is abundantly apparent that the TM-O and O redox events are differentiated. The key point in the manuscript is the link between TM migration and redox. However, one might pose a "chicken or egg" question. Is it transition metal migration that's ultimately supporting anionic redox, or could it be that the oxygen was already reduced, destabilizing bonding and enhancing transition metal mobility? Further, the mechanism notation appears overly simplistic.
2. The authors provide the XRD refinement data to support that TM migration in the bulk is reversible during charging and discharging. While there is clear evidence that TM migration is irreversible at least on the subsurface region. The author needs to provide explanation in detail why TM tends to migrate back to Li layer during the discharging process. XRD is not the best tool for this purpose by the way
3. In terms of transient states, a few critical pieces of information are lacking: (i) In spite of the authors focus, barriers for TM mobility are still critical factors, and if literature values can't be cited for at least similar transitions, the calculations should be performed; (ii) Lacks correlation between experimental and computational state of charge;
4. All the STXM mapping shown in this manuscript is 2D average information. How can the authors confirm the oxygen redox shuffling occurs in the bulk instead of only on the surface, given that transmission mode includes both bulk and surface signal.
5. Notation on the computational model bonding is utterly confusing; similarly with the diagrams. Perhaps a side view of with the transition metal migration with oxygen sites labeled would help.

Reviewer #2 (Remarks to the Author):

The manuscript has been well updated on the basis of reviewer comments, and now this manuscript is suitable for Nature Commun.

Reviewer #3 (Remarks to the Author):

In this new version of their manuscript, Gent et al. have carefully and convincingly answered to the major criticisms concerning their paper.

In particular, from the point of view of data analysis methodology, the new added information substantiate and globally supports the proposed interpretations much better than in the previous

version.

I cannot find any major flaw in the proposed interpretation, and find the paper very interesting and suitable for publication in Nature Communication.

Reviewer #1

1. It is abundantly apparent that the TM-O and O redox events are differentiated. The key point in the manuscript is the link between TM migration and redox. However, one might pose a “chicken or egg” question. Is it transition metal migration that’s ultimately supporting anionic redox, or could it be that the oxygen was already reduced, destabilizing bonding and enhancing transition metal mobility?

This is an interesting question which we have been investigating in a separate work. Based on the present data, while we confidently conclude that there is a correlation between O redox and TM migration, we cannot say which one precedes the other. Answering this question is beyond the scope of this manuscript, as it requires substantial computational and experimental work to probe the intermediate states. Some computational studies have suggested that TM migration in Li-rich materials is promoted by oxidized oxygen ions (J.-M. Lim, D. Kim, Y.-G. Lim, M.-S. Park, Y.-J. Kim, M. Cho, K. Cho, *J. Mater. Chem. A* **2015**, *3*, 7066). We have included the following discussion in the main text to address this point:

Further work will seek to understand the relative kinetics of O redox and TM migration, identify the intermediate states, and establish whether, for example, O_{2p} states are first depopulated, promoting TM migration through (de)stabilization of specific structural configurations, as has been predicted computationally.⁶⁹

Further, the mechanism notation appears overly simplistic.

We have included the following discussion in the main text, where the equation is first proposed in the discussion section:

In the specific case where a nominally 4+ Ni ion migrates into an octahedral Li vacancy in the Li layer, this can be written using Kröger-Vink notation⁷⁰ as: $Ni_{Ni}^{\times} + O_O^{\times} + V_{Li}' \rightarrow Ni_{Li}^{'''} + V_{Ni}^{''''} + O_O^{\cdot} + e^{-}$. Since many different migration pathways for the various TMs are possible, we represent the substantial change and distribution in the oxygen coordination environments with $\{O^{-} + TM_{mig}\}$, which modulates the oxygen redox potential and shifts a large fraction to lower voltage (< 3.65 V) after the first charge.

2. The authors provide the XRD refinement data to support that TM migration in the bulk is reversible during charging and discharging. While there is clear evidence that TM migration is irreversible at least on the subsurface region. The author needs to provide explanation in detail why TM tends to migrate back to Li layer during the discharging process. XRD is not the best tool for this purpose by the way

In addition to the existing references citing previous literature in agreement with this observation (N. Yabuuchi *et al.* *J. Am. Chem. Soc.* 2011, *133*, 4404; I. Takahashi *et al.* *J. Phys. Chem. C* 2016, *120*, 27109; C. R. Fell *et al.* *Chem. Mater.* 2013, *25*, 1621; N. Ishida *et al.* *Journal of Power Sources* 2016, *319*, 255; D. Mohanty *et al.* *Nano Energy* 2017, *36*, 76.), we have moved the existing discussion to the first paragraph of the Results section, where the XRD results are

first presented, and included more detailed explanation on the plausibility of partially-reversible TM migration:

We note that this intra-cycle partially reversible bulk TM migration, which has been observed previously,^{34-36,40,43} is distinct from the permanent TM migration that occurs at the surface during the first cycle, which is known to occur as a result of oxygen evolution and densification,^{27,49-51} and from the longer-term irreversible TM migration associated with the formation of a spinel-like structure and voltage fade.⁵² Reversible TM migration is consistent with the TM site preference varying with electron count and/or Li content,⁵³ such that on charge certain electronic and structural conditions are met that promote TM migration, while on discharge these conditions are quenched by lithiation and the original TM sites become favored again. The partial irreversibility (i.e. the resulting in-plane disorder and residual TM occupancy in the Li layer) suggests some “trapping” of TM ions as well as hysteresis in the migration pathways during charge and discharge.

3. In terms of transient states, a few critical pieces of information are lacking: (i) In spite of the authors focus, barriers for TM mobility are still critical factors, and if literature values can't be cited for at least similar transitions, the calculations should be performed;

Reasonable TM migration barriers (as low as 0.1 eV) for similar migration pathways in delithiated Li_2MnO_3 have been calculated previously as a function of Li content (E. Lee, K. A. Persson, *Adv. Energy Mater.* **2014**, *4*, 1400498; J.-M. Lim, D. Kim, Y.-G. Lim, M.-S. Park, Y.-J. Kim, M. Cho, K. Cho, *J. Mater. Chem. A* **2015**, *3*, 7066).

We have included the following sentence in the discussion of the computational results to more clearly address the migration barrier concern:

Similar migration pathways have been shown to have reasonable migration barriers (as low as 0.1 eV) in Li_2MnO_3 .^{68,69}

(ii) Lacks correlation between experimental and computational state of charge;

We included in the revised Supplementary Information (Supplementary Figure 9) calculations with non-zero Li content, at a composition of $\text{Li}_{0.17}\text{TM}_{0.83}\text{O}_2$, corresponding to a capacity approximately equal to the end of the plateau where we do experimentally observe TM migration. The same phenomenon is observed, even when Li coordinates the affected oxygens. Others (E. Lee, K. A. Persson, *Adv. Energy Mater.* **2014**, *4*, 1400498; J.-M. Lim, D. Kim, Y.-G. Lim, M.-S. Park, Y.-J. Kim, M. Cho, K. Cho, *J. Mater. Chem. A* **2015**, *3*, 7066) have computationally predicted TM migration at a variety of Li contents in Li_2MnO_3 .

4. All the STXM mapping shown in this manuscript is 2D average information. How can the authors confirm the oxygen redox shuffling occurs in the bulk instead of only on the surface, given that transmission mode includes both bulk and surface signal.

We have included the following discussion in the main text to clarify this point:

Although all pixels measured with STXM do contain some surface signal, the surface contribution decreases with particle thickness, being lowest in the center of the particles' 2D projections. Since the ellipsoidal LMR-NMC particles are several hundreds of nanometers in thickness in the center, contribution from the surface (a few nanometers) is small, and the signal is dominated by the bulk. For the same reason, near the edge of the particles, surface signal dominates.

5. Notation on the computational model bonding is utterly confusing; similarly with the diagrams. Perhaps a side view of with the transition metal migration with oxygen sites labeled would help.

We appreciate the reviewer's comments and have modified Figure 6 and Supplementary Figure 9 accordingly:

Figure 6. Effect of TM Migration on Electronic Structure. (a) Plot of the O fractional oxidation state (red) and the migrated TM fraction (green) as a function of capacity, showing the clear link between hysteresis in the TM migration and voltage hysteresis in the O redox. Error bars indicating fitting residual and refinement error for the O oxidation state and TM migration fraction, respectively, are smaller than the data symbols and are therefore not shown. (b) pDOS for the TMs and the two-coordinate (O⁽¹⁾, blue) and three-coordinate (O⁽²⁾, black) oxygen environments in the pristine delithiated state (top), and after Mn (middle) and Ni (bottom) migration into octahedral sites in the Li layer. The integrated charge density for the lowest unoccupied states in the pristine delithiated structure (blue shaded area of the pristine DOS) is shown in the top right inset. The total TM pDOS is normalized by the number of TMs in the supercell for comparison with the single-site oxygen pDOS. Schematics of each supercell used to generate the pDOS

are shown to the left, with the plotted O⁽¹⁾ and O⁽²⁾ oxygen environments circled. (d) Schematic of the reorganization of the electronic structure due to TM migration.

Supplementary Figure 9. DOS of other Migration Pathways. (a) DOS of a partially delithiated supercell before and after migration of Ni and Mn, with the decoordinated $O^{(1)}$ becoming additionally coordinated by a Li ion after TM migration. The results are largely similar to those of the fully delithiated supercell. (b) DOS of a fully delithiated Mn-only supercell before and after migration in O3 and O1 stacking configurations (see Methods). The shift of the $O^{(1)}$ states to higher energy is observed in both cases, though appears to be less than in the mixed Ni and Mn supercell.

REVIEWERS' COMMENTS:

Reviewer #4 (Remarks to the Author):

From the extremely detailed referee comments and authors' responses, it is clear that this paper addresses a very complex problem with a very active research community. While I do not find every answer from the authors to be completely convincing, I do think they have understood the essential comments from all referees and have made a significant attempt to answer the important concerns. I think the paper now represents a much clearer representation of the authors' data and their hypotheses, and I think the paper should be published in Nature Commun. I think this paper will generate a large amount of interest and impact in the battery community.